# Investigating the experiences of low-carbohydrate diets for people living with Type 2 Diabetes: A thematic analysis

Lisa Newson*, Francesca Helen Parody¤

School of Psychology, Faculty of Health, Liverpool John Moores University, Liverpool, United Kingdom

¤ Current address: The Wellcome Trust, London, United Kingdom
* l.m.newson@ljmu.ac.uk

## Abstract

Low-Carbohydrate Diets (LCDs) are becoming increasingly popular to manage Type 2 diabetes mellitus (T2DM). However, there is a need to identify people with T2DM's understanding of LCDs, their reasons for engaging in such a diet, how they sustain it and any challenges they face. This study aimed to assess the experience of adhering to a LCD within a sample of individuals with T2DM. Ten participants with T2DM were recruited using a self-selecting sampling method from an online diabetes community that promotes LCDs. Participants completed one-to-one digitally recorded semi-structured interviews, which were later transcribed verbatim and data subjected to Thematic Analysis. Five core themes and twelve subthemes were developed during the analysis: (1) Lack of professional guidance; (2) Fear of complications & long-term medication use; (3) Dietary control as motivation; (4) Positive health outcomes; and (5) Social support. The findings are discussed with reference to a psychological model of behaviour, *COM*-B. Participants reported gaining knowledge and skills to increase their *C*apability to engage in LCDs, *M*otivation to manage diabetes outcomes influenced adherence. However, challenges were reported with the *O*pportunity to engage in behaviour, mainly influenced by social support. Health professionals and significant others may benefit from resources to help build knowledge and understanding and assist with maintaining a LCD long-term.

## Introduction

Diabetes Mellitus (DM) is a worldwide health problem; in 2019, 463 million adults were estimated to have DM, and this is projected to increase to 700 million by 2045 [1, 2]. Type 2 Diabetes Mellitus (T2DM) accounts for 90% of all diabetes cases in the UK [3]. T2DM is related to insulin resistance and relative insulin deficiency [4] and can cause serious complications, including heart disease, stroke and kidney failure [5]. Diabetes Mellitus is also extremely expensive; it accounts for 10% of the UK National Health Service (NHS) budget, with 80% of this due to complications, and these expenditures are projected to increase [6]. In light of these worrying statistics, it is critical that studies focus on ways to reduce prevalence.

**Data Availability Statement:** Data cannot be shared publicly because verbatim full transcript release has not received ethical approval or participant consent. However, as per ethical approval; raw data has been included as evidence

via extracted quotes from verbatim transcripts as samples of evidence. The authors confirm that the data supporting the findings of this study are available within the article. Data are available from the Liverpool John Moores University, School of Psychology Ethics Committee (contact via l.m. newson@ljmu.ac.uk corresponding author or School of Psychology Ethics Committee PSYREPsubmissions@ljmu.ac.uk) for researchers who meet the criteria for access to confidential data.

**Funding:** The authors received no specific funding for this work.

**Competing interests:** The authors have declared that no competing interests exist.

Dietary modification is considered an integral aspect of effective T2DM self-management and can exert effects through multiple metabolic and physiological pathways. For instance, as well as leading to weight loss and lowered blood glucose levels [7], diet can also impact blood lipids, blood pressure and systemic inflammation, influencing cardiovascular risk factors [8]. In addition, research has even suggested that remission of T2DM may be achieved through the use of specific diets [9]. However, there is mixed evidence regarding the best overall dietary approach for T2DM.

Since the 1950s, the American Diabetes Association (ADA) has provided national standards for dietary care for diabetes. Although its early recommendations included high levels of carbohydrates (ranging from 45 to 60% of caloric intake), for years, its guidelines have not specified an ideal distribution of macronutrients among carbohydrates, fats and proteins [10]. It also recognises the importance of nutrition therapy in diabetes management and encourages patients to work alongside their health providers to generate an individualized meal plan [11]. In comparison, the UK National Institute for Health and Care Excellence (NICE) guidelines [12] advise patients with T2DM to eat starchy carbohydrates, fruit and vegetables and lower their fat intake. Information on healthy diets for the general public on the UK National Health Service (NHS) website [13] promotes adherence to the *Eatwell Guide and states* 'that just over a third of your diet' should 'base meals on higher fibre starchy foods like potatoes, bread, rice or pasta'. A significant amount of research has been dedicated to investigating specific diets in relation to T2DM management. For example, the Diabetes Remission Clinical Trial (DiRECT) focused on a very low-calorie diet; at 12 months, it found that 46% of participants achieved T2D remission and stopped taking antidiabetic drugs. However, it should be noted that healthcare professionals delivered the intervention entirely within routine primary care. Thus, maintaining patient weight loss may be difficult when primary care support ceases, and patients have reduced professional and social support and revert to old habits that started their weight gain [14]. Alternatively, literature has provided support for a Palaeolithic diet, consisting of mainly meat, vegetables, fruits and nuts; Masharani and colleagues [15] found that patients with T2DM demonstrated improvements in glucose control and lipid profiles. Dietary Intermittent fasting has also received attention in relation to T2DM and has been found to improve body weight and fasting glucose levels [16]. However, larger scale, longitudinal studies are required to strengthen these findings.

Specifically, interest in Low-Carbohydrate Diets (LCDs) for T2DM management has increased among experts, clinicians and the public in recent years [17]. Low-Carbohydrate (LC) approaches originate from the idea that lowering the fat-storage hormone insulin improves cardio-metabolic function and promotes weight loss [18]. LCDs reduce the total daily intake of carbohydrates whilst increasing the consumption of fat and ensuring sufficient protein levels. There is considerable debate surrounding what is defined as a LCD. Eenfeldt [17] has specified three differing variations: ketogenic low carb (<20 gram carbs per day), moderate-low carb (20–50 gram carbs per day) and liberal low carb (50–100 gram carbs per day). A practical guide [19] "adapting diabetes medication for low carbohydrate management of type 2 diabetes" was published for primary care physicians and defines a LCD consisting of <130 g of digestible carbohydrates per day [20]. Indicating a liberal LCD as opposed to a much more restricted Ketogenic approach [20].

Research has provided support for the effectiveness of LCDs for T2DM, in particular through reducing blood glucose levels, body weight and improving systolic blood pressure and HDL cholesterol levels [20–24]. In addition, evidence suggests that LCDs are superior to other diets, such as low-fat diets, in regulating blood lipid levels and reducing body mass index and insulin dose in T2DM patients [25].

However, despite evidence of its effectiveness for the treatment of T2DM, many health professionals are not recommending LCDs to patients. A study [2] found that 48% of dietitians advised patients to only occasionally or frequently reduce their carbohydrate intake, and over a third considered 30–39% carbohydrate of total daily energy intake as realistic. LCDs have created controversy in healthcare [26]. There is an ongoing debate surrounding the impact of dietary fat on cardio-metabolic health. A reduction in total and saturated fats has been advocated consistently in nutritional guidelines over the years, and studies have argued that such a diet may reduce one's risk of cardiovascular disease (CVD) [27]. On the other hand, the multinational Prospective Urban Rural Epidemiology (PURE) study [28] reported that total fat is not associated with cardiovascular disease, myocardial infarction or cardiovascular disease mortality, but, unexpectedly, with lower all-cause mortality. Furthermore, a panel of experts organized by the British Medical Journal concluded that it is meaningless to consider the impact of total fat alone and that all types of fats must be taken into account [29].

The growing popularity of the LCD movement has seen guidelines from professional groups promoting carbohydrate restriction as a viable treatment option. For instance, an updated report during 2020 by the ADA [30] has stated that reducing carbohydrate intake has demonstrated the most evidence for improving glycaemia in individuals with DM. Moreover, it is noteworthy that organisations such as Diabetes Australia [31], Diabetes UK [32], and the American Diabetes Association in conjunction with the European Association for the Study of Diabetes [33], have released position statements and specific recommendations regarding LCD's for people with diabetes.

Regardless of the dietary approach adopted, many patients with T2DM report difficulties maintaining specific dietary regimes, noting that many psychosocial, behavioural and environmental influences affect eating habits [34]. For example, a qualitative study found that patients with T2DM felt they lacked aspects of dietary knowledge and found specific diet details confusing, which may have impacted their self-efficacy and thus their diet adherence [35]. In addition, Vijan et al. [36] noted that social support, particularly from family members, was highly influential on a patients' ability to follow dietary regimes. Patients also presented emotional struggles associated with their diet, difficulties during social occasions, and a dislike of the foods. Similar concerns have been raised surrounding the long-term application of LCDs, which have suggested poor adherence to the LCDs beyond six months (particularly for ketogenic diets, very low carb diets) [37]. However, it is noteworthy that people may find any dietary change challenging. Despite some controversy, the LCD approach to T2D management has increased support and more patients across the world are either being supported by health professionals (for example a Low Carb programme has been endorsed and commissioned by some NHS services [38]) or are independently choosing to follow this approach (without health professional endorsement).

Although some research has examined the quantitative influence of LCDs on patients with T2DM, very little attention has been given to qualitative research on LCDs. There is a lack of real-world research investigating individuals who have followed their own LCD compared to diets that have been prescribed for study trials. Due to the increasing popularity of LCDs for T2DM in recent years, research should focus on why patients adopt this dietary stance and how they manage and sustain it. In 2019, one mixed-method study conducted in South Africa followed twenty-eight participants and evaluated dietary assessment, T2DM clinical outcomes (e.g. HbA1c changes/ diabetes remission status), and interviewed participants about their diet experiences [39]. However, further research is required to explore the subjective perspectives of participants with T2DM who adopt a LCD.

The current study investigated the experiences of LCDs for people with T2DM. This research is the first to utilise a psychological investigation to understand lay people's

**Table 1. Participant demographics.**

| Respondent | Sex | Age (years) | Time since diagnosis | Time since starting LCD | Average carbs consumed per day (grams) |
|---|---|---|---|---|---|
| MG | Male | 58 | 4 y 9 m | 4 y 9 m | 10 |
| S | Female | 56 | 4 y 11 m | 4 y 11 m | 20 |
| Y | Female | 54 | 3 y 7 m | 3 y 6 m | 10 |
| C | Female | 73 | 5 y | 4 y 6 m | 20–30 |
| DS | Female | 57 | 2 y 7 m | 2 y 7 m | 30 |
| MT | Male | 59 | 23 y | 5 m | 30–40 |
| CW | Female | 56 | 9 y | 9 y | 50 |
| DC | Female | 62 | 3 y | 2 y 6 m | <20 |
| M | Female | 64 | 5 y 2 m | 2 y 10 m | <30 |
| JW | Male | 72 | 6 y 7 m | 7 m | 30 |

y, years; m, months.

perceptions of LCDs and their reasoning and motivations for engaging in such a diet. In addition, a key aspect of this research focused on how people can sustain their LCD, how they manage it, and any barriers or challenges they face. Finally, the project also explored the perceived effects of LCDs on people's T2DM and their general health and wellbeing.

## Method

### Design

Qualitative methods have proven highly valuable within nutrition and dietetics research [40]. A Reflexive Thematic Analysis (TA) [41, 42] was considered suitable for this study. Semi-structured one-to-one interviews were conducted in order to gather the subjective experiences of patients living with T2DM and following a LCD.

### Participants

Following ethical approval from Liverpool John Moores University (PsyREP, School of Psychology Ethics Committee, 02/03/2020). Gatekeeper consent to promote this research study was sought from a UK online health platform and community support network (membership 300k plus), which actively promoted LCD to those with and without T2DM. An advert was placed on the research section of their forum seeking participants living with T2DM to take part in an online interview about their experiences and maintenance of LCDs. Ten male (30%) and female (70%) participants with T2DM self-selected for participation in the study (Table 1).

Participants were required to meet the inclusion and exclusion criteria (Table 2). The participants had been living with T2DM for a mean of 6.75 years (ranging from 2.5 years to 23 years) and were aged between 54–73 years old (mean = 61). Most had been maintaining the application of a LCD for a significant period of time (mean 3.5 years, range 5months- to nine years), mostly following a moderate-LCD approach.

**Table 2. Study inclusion and exclusion criteria.**

| Inclusion criteria | Exclusion criteria |
|---|---|
| • Aged 18 +<br>• Have had a T2DM diagnosis for $\geq$ 12 months<br>• Have been on a LCD (consuming $\leq$ 100 grams of carbs per day) for $\geq$ 3 months | • Poor understanding of the English language<br>• Experienced any recent changes in diabetes medication<br>• Had any additional conditions (e.g. Type 1 diabetes, coronary heart disease, mental illness) |

## Data collection

The second author (FHP) conducted one-to-one semi-structured interviews during March-April, 2020. The participants received an information sheet, consent information and a health-screening questionnaire to read and process before to the interview. Neither author had any prior relationships with the study participants. Therefore, upon commencement of the interview, the researcher talked the participants through the procedure, built an initial rapport (to reduce any anxieties), and provided an opportunity for participants to ask questions before the recording commenced. Participants were reminded that they were free to withdraw from the study, that their personal information would remain confidential, but that selected quotes from their raw data transcripts may be used in subsequent publications.

Semi-structured interviews allowed for detailed and personal discussions surrounding the participants' experience of LCDs. The use of open-ended questions such as "How have you found maintaining your LCD?" gave the participants the opportunity to discuss aspects important to them. This also allowed the researcher to follow up any themes of interest that developed from the discussions. An interview schedule was used as a basis to guide the semi-structured interviews. The first part of the interview addressed why the participants decided to engage in a LCD (e.g., "What are your attitudes, thoughts and feelings towards low carb diets?"). The second part explored how participants managed and sustained a LCD (e.g., "What aspects of this diet do you find easy or difficult?"). The format of the questions was informed initially by the Theory of Planned Behaviour [43] in order to gather theory-grounded data (although this theory was not used to analyse the data, as an inductive stance was applied). Prompt questions were also included to ensure that the interview imitated a 'flowing conversation' [44].

The interviews were recorded via an online video platform, with a mean duration of 36 minutes (range 20–65 minutes). The recordings of the interviews were transcribed verbatim to create raw data files (and subsequently, audio recordings were deleted). After the interview had concluded, the participants were debriefed and thanked for their participation.

## Data analysis

Reflexive Thematic Analysis (TA) [41, 42] was applied to identify, analyse, and report patterns and themes within the interview data. TA was considered appropriate for the current study as it can provide a rich and insightful yet complex account of the data. TA is useful for examining people's views and perceptions, which we identified as important for understanding people's experiences of a LCDs [42]. Furthermore, the data were analysed inductively; thus, it was not driven by pre-existing coding frames or the researcher's analytic preconceptions. TA consisted of various phases of data familiarization, coding and thematic development [42].

## Methodological integrity and researcher description

To promote quality within the research study and in the writing of this article, we applied, where appropriate, the Consolidated Criteria for Reporting Qualitative Research (COREQ) checklist [45]. Evaluative criteria were also used to ensure quality and trustworthiness in the current study. The data was read and re-read multiple times by FHP, who also developed initial codes and themes. FHP developed a reflective report during this process to demonstrate self-awareness in the analytical process and enhance qualitative researcher skills. Through researcher discussion, themes were revisited, and interpretations were worked-up further to provide in-depth analysis. Thick description was used to address transferability. The context of the research, such as the setting, sample size, and participant characteristics, was noted to allow others to judge its applicability to other settings accurately.

Researcher triangulation [46] promoted objectivity between the researcher's position and the analysis. The first author (LN) was a White female Registered Health Psychologist, Reader in Applied Health Psychology (D.Health Psyc), had research interests in diabetes and expertise in qualitative research and second author (FHP) a White female health psychology trainee (BSc, MSc) interested in low carb diets and diabetes research. Discussions occurred between authors throughout the study process to ensure a shared understanding and agreement.

Direct quotes (raw data extracts) from a range of participants act within this article as evidence to support commentary. However, due to the nature of this qualitative research, in line with legal and ethical processes, participants of this study did not agree for their full transcripts to be shared publicly, so supporting data beyond the sample quotation extracts is not feasible.

## Findings

Five core themes and twelve subthemes were developed during the analysis. The core themes included: (1) Lack of professional guidance; (2) Fear of complications & long-term medication use; (3) Dietary control as motivation; (4) Positive health outcomes; and (5) Social support. Verbatim quotes have been provided as evidence of such. Fig 1 displays these results and demonstrates how the themes interlink.

### Theme 1: Lack of professional guidance

Participants were mostly unsatisfied with the guidance they received from their health professionals (e.g. doctors) regarding T2DM management and specifically suggested that dietary advice was vague and insufficient.

> . . .the diet stuff was absolutely terrible, it was just, "Eat healthfully". No indication of what healthfully was. (DC, P4, 121–123)

Many participants noted the standard UK dietary advice, which they considered lacked personalization and relevance to their T2DM and their individual behavioural needs.

> I was given a photocopy of the EatWell Guide, which must have been photocopied 97 million times // I mean, it's almost unreadable anyway, as well as being complete rubbish. (MG, P3, 95–100)

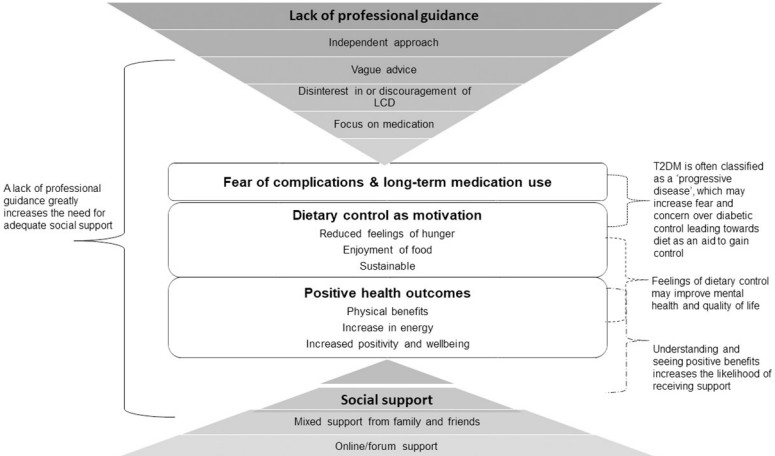

**Fig 1. Relationship between key themes.**

Participants reported a sense of feeling lost, with limited opportunity to explore dietary options or variations in approach; participants typically reported a lack of guidance and support in developing the skills required to change their dietary behaviour.

> I was confused because I didn't know what to do. I was being asked to go on a diet that I really felt was illogical. . . (JW, P6, 212–214)

Some participants claimed that the first (and occasionally only) line of treatment offered to them was medication rather than lifestyle guidance.

> They don't tell you about any (low carb) diet; they just put you automatically on medication. (DS, P3, 91–93)

The medication pathway was viewed as a "mop and bucket" (DS) approach to T2DM management, rather than a viable long-term solution to personalize their intervention and tailor their T2DM treatment. As a result, several participants rejected medication from their healthcare professional or stopped taking it after a short time. Respondent MT, in particular, expressed frustration (and possible anger) towards his doctor for the lack of consideration of a personalised 'holistic approach' he was seeking.

> His solution was to get me on Metformin and statins. I was on both of those in the past. The Metformin gave me terrible stomach cramps and just couldn't tolerate it //, but that was their solution, to just give me more pills. (MT, P4, 118–126)

In light of the above, it is perhaps not surprising that all participants reported that they had initiated and developed their knowledge and skills to implement a LCD independently. Many emphasized the importance of an "evidence-based approach" (MT) and thus consulted books and online materials to increase their understanding of the LCD. Some participants were able to use previous research experience to aid this process.

> I have a scientific degree, and I started reading scientific research articles // after time, I found more and more resources online about eating low carb. (S, P2, 44–55)

In some instances, participants asked their healthcare professional to give them time to explore alternative options available. Many agreed that this led them to the online diabetes website, which was a primary source of support and information.

> I said, "Can you leave it with me" to the diabetic nurse. I went home, and I Googled 'reversing diabetes', and it took me to [named website]. (DS, P3, 96–99)

The majority of the participants indicated that they had informed their healthcare professional about their LCD. Responses ranged from discouragement: "not good, especially not in the long term" (DS), "that's rubbish" (MG), to disinterest: "they always shut their ears" (M), "he just listened and did not comment on it" (S). Such comments were deemed common responses, given "these types of comments are typical- as others say on the LCD forum" (DS).

In rare cases, respondents reported that their doctor supported their diet, which was welcomed and reinforced motivation to engage in an attempt to take control of their T2DM.

Even when I went back to her and said, "No I haven't tried the statins or the Metformin" she (doctor) just accepted that and was still supportive, and when she saw the improvement in my HbA1c she said, "Fantastic, you've done a great job." (MT, P18, 705–712)

The importance of social support in promoting knowledge and skills to engage in a LCD was recognised by all participants. For example, as a result of engaging with the LCD, one participant, DS, described how they were working alongside healthcare professionals to bring awareness of LCDs for T2DM and offered direct support to other patients.

I work (volunteer) with five surgeries now locally. So somebody that's been diagnosed comes in. I can sit and talk to them for an hour, whatever it is, and just go through it and maybe help them if they've got any concerns. (DS, P11, 454–458)

## Theme 2: Fear of complications and long-term use of medication

Many participants stated that they were "frightened" (MG) and "afraid" (JW) of living with uncontrolled diabetes as well as experiencing devastating complications. This fear motivated them to initiate and maintain their LCD. In addition, some respondents had witnessed family members suffer from T2DM (e.g. S), which encouraged them to address their illness. Moreover, some feared becoming a "burden" (DS) to their families or being unable to experience a usual way of life.

My overriding motivation is to not have a heart attack, be around for my kids, and enjoy my life. (CW, P5, 196–197)

Distress over long-term medication use was another motivating factor highlighted by participants. Many wanted to avoid relying on medication, particularly insulin.

. . .my GP said, "Well, you know, there wouldn't be any way around insulin", and I really didn't want to start insulin. (S, P3, 108–110)

. . .the motivation is (to manage) my glucose control at a level that is substantially away from me to have to take drugs for my diabetes. (JW, P5, 191–193)

Participant Y expressed that they refused to take medication unless they could see a future in which it was no longer necessary. Participants expressed a desire to be in control of their health and management of T2DM.

I wasn't averse to Metformin, which was the medication being discussed, so long as I could see what the pathway would be to getting off it. At that point, she said, "You probably wouldn't get off it." So at that point, I lost interest in discussing it any further. (Y, P5, 163–169)

## Theme 3: Dietary control as motivation

Many of the participants acknowledged a reduction in hunger since following their LCD. Respondent DS described it as a "switch" that has been "turned off" (DS). This was often compared against other diets in which they felt they were constantly hungry. Participant M spoke of their previous diet:

I was eating quite a lot of carbs, but not that many calories I don't think, so I was hungry all the time. (M, P2, 60–62)

Some participants felt they were no longer receiving "triggers" (MG) for hunger until later in the day, which resulted in behaviour changes such as less frequent meals and less snacking. Respondent DC highlighted how they were more in tune with their body and only ate when they felt necessary:

I don't need to eat unless I'm hungry, which might be twice a day or it might be once a day, it might be more, but I don't snack or anything else. (DC, P11, 461-

465)

A frequent comment made by participants was that they thoroughly enjoyed the food they were eating on their LCD. They did not feel as if they were "depriving" (CW) themselves and were able to eat foods they had previously always denied.

I could have cream in my coffee. I could have cheese, bacon, just lovely things. . .(DC, P5, 168–173)

This also made it easier for some participants to sustain their LCD long-term.

I haven't gone off it at all. I enjoy the food too much. (C, P5, 166–167)

Indeed, most of the participants viewed their LCD as being "very sustainable" (DS). Although reducing carbohydrate intake was considered "difficult initially" (Y), it became easier to maintain over time. In addition, several participants, such as MG, stated that they did not miss carbohydrates "at all". Some participants referred to their LCD as a lifestyle rather than a diet.

LCHF [low-carb, high-fat] is not a diet; it is a lifestyle. A diet implies that it's something that's going to end at some point // I don't see that that will happen. I see that I will continue to eat this way for as long as I'm on this planet. (MT, P10, 385–390)

A few participants believed the key to successfully sustaining their diet was to be kind to themselves and to allow for flexibility.

I'm not attempting to have unrealistic expectations about my ability to do this stuff. // if I occasionally fall off the wagon, I'm not going beat myself up over it. (MT, P15, 567–571)

## Theme 4: Positive health outcomes

All the participants claimed that their LCD had improved their T2DM. All respondents reported that their HbA1c levels had dropped significantly f, and some had managed to put their diabetes into remission. According to DC "everything is now at completely normal, optimal levels". The majority of participants also reported weight loss, which came as a pleasant surprise.

I didn't think I would lose weight because I've tried dieting before //, but when I started losing weight I thought, this works! (C, P3, 81–85)

A wide variety of other physical benefits were also discussed during the interviews. Respondent MG said:

> I used to have really bad acid reflux, that's gone. I used to have sleep apnoea which has gone. I used to snore really badly, which I don't do quite as much. . . (MG, P6, 223–227)

On the other hand, one participant stressed that they had been experiencing multiple unpleasant side effects since engaging in a keto diet, which were yet to cease.

> . . .for a little while, I felt weak physically, and I had to reduce my exercising. I've had periods where I've felt sick every day for 2–3 days. (JW, P8, 299–301)

In addition, a common reflection made by participants was that they felt more energetic on their LCD. This increase in energy levels allowed them to exercise and accomplish daily tasks more easily.

> I've been a sporty person and played sports throughout my life, but now I just have huge reserves of energy. (Y, P11, 440–442)

As a result of these positive changes, many participants described an increased enjoyment for life and improvements in their mental health, such as gaining "confidence" (DS) and feeling more "resilient and calm" (M). Participant S adds: "I probably feel healthier than the average person my age". This increase in well-being acts as a strong motivator for many respondents to continue their LCD into the future.

> I want to carry on doing it because I feel so much better. Everything in my life has gotten better since I did it. (MG, P8, 313–315)

## Theme 5: Social support

A potential challenge when sustaining a LCD was the opportunity to select a low-carb based meal when eating out at restaurants. Adjusting to external influences on food choices was deemed challenging, participants acknowledged this as especially noteworthy during early adoption of the diet, and having the confidence to speak out and make a dietary request, and in doing so, to also not feel judged by others (who perhaps didn't understand or agree with their LCD choice).

> I did find it tricky, to begin with, and I felt a bit awkward asking to not have this, not have that // I don't think about it now I just ask and they say, "Yeah, that's fine." (DC, P12, 488–493)

Participants reported mixed feedback regarding the quantity and quality of social support received regarding their LCD. The majority of participants in this study had been adhering to LCD's for some time (mean 3.5years) as such, those who were successful acknowledged how their families were "proud" (DC) of them and that they were supportive of their dietary choice after witnessing the difference it had made to their health.

> My husband is very supportive because he has also seen the results. He has accepted that it's the way I have to eat. (S, P6, 227–229)

Upon reflection, others stated that they had to "educate" (M) their families in order to receive support, whereas some respondents believed their families would simply "tolerate" (JW) their diet. Occasionally participants would convey frustration at their family's lack of understanding or consideration of their LCD:

> People like my mother-in-law. She's always trying to encourage me to have a dessert // and even after all this time. It makes me quite cross because I don't eat dessert. (CW, P7, 276–280)

Some family members also expressed worry or concern over the types of food (e.g. high fat, MG) respondents were eating. Consequently, a lack of approval, or a challenging conversation, or a fear of "upsetting" (M) others may deter people from engaging in or being able to maintain a LCD.

> . . .if it's not endorsed by your friends and family or the people you mingle with, that makes it quite hard to keep going, and you also think, *is this the right thing to do*? (CW, P11, 423–426)

Consequently, all participants reported that they had used websites and online support groups to seek information and advice whilst following their LCD. This was seen as highly beneficial, particularly as a way to connect with others who had been through a similar experience. This social support was deemed education, helping them to learn about LCD and how to implement the diet into their daily lives, but it also provided an ongoing opportunity to seek clarification and reinforcement.

> It's good to learn from people who've been there and done it, who might have experienced the same things as you and who can provide advice. (Y, P5, 190–193)

Online support, particularly the LCD diabetes forum, was considered a key part of the participants' LC journeys. According to DC: "I don't think I would be in the place I am now if it wasn't for them." Respondent M also said they needed their "handheld" initially after following a low-fat diet for many years and found the online LC community to be very supportive. Sufficient support was seen as particularly essential immediately after diagnosis.

> I think it's really important as I say, because it's a shock, the diagnosis is a shock. (CW, P9, 345–347)

## Discussion

To our knowledge, this is one of the first studies to qualitatively examine the experience of individuals with T2DM adhering to a self-prescribed LCD. The researchers were particularly interested in how and why participants started their diets, the ease and means of sustainability, and any challenges that may have been encountered.

Whist the study methodology and analysis has been conducted inductively, it is beneficial to consider the findings utilising a psychological approach. Specifically, the COM-B model [47], which stands for Capability (C), Opportunity (O), Motivation (M) and Behaviour (B), and suggests that changes in these components (COM) influence behavioural outcome (B). In this study, the behaviour (B) was the implementation of a LCD, and the analysis suggests that

these participants were able to demonstrate successful changes in their capability, opportunities and motivations, hence demonstrating a mean application of the LCD for 3.5 years.

In this study, participants became capable, mainly through their own evidence-seeking behaviour and skilling up. Participants were able to seek out information, became knowledgeable on LCD's, and made informed decisions to follow a LCD, intending to gain control over their T2DM. None of the participants had been prescribed or recommended a LCD by a healthcare professional. Instead, they started their diets independently, with the majority using books and online materials for guidance. This is considered controversial, as researchers have argued that to implement a LCD safely and effectively, people with T2DM should receive medical supervision and behaviour modification support where possible [48]. Overall, participants demonstrated a coherent understanding of basic T2DM knowledge and had all thoroughly researched LCD's.

Participants were highly motivated. Specifically, their reflective motivation was managed by their T2DM goals and intentions to seek control over their health. Their automatic motivation was reinforced through improvements in their health status (e.g. T2DM outcomes) and by reinforcement received from significant others (or those within the LCD social networks). Many feared the prospect of uncontrolled T2DM and suffering from chronic T2DM complications such as retinopathy or myocardial infarction. Long-term reliance on diabetes medication was also a source of concern. This strongly motivated several participants to initiate and sustain their LCD. All of the participants spoke about how they had regained some sort of dietary control since following their LCD. A reduction in hunger was commonly mentioned, mirroring research by Kelly, Unwin and Finucane [26], who stated that many of their patients reported reduced hunger and increased satiety after engaging in a LCD. In contrast with Vijan et al.'s [36] findings, participants considered the food on their LCD to be enjoyable and satisfying, which helped them sustain their diet. All but one of the participants had maintained their diet for over six months, with a mean of 3.5 years and the participants described LCD as a lifestyle and not a diet (with an endpoint). These findings support research by Cucuzzella, Tondt, Dockter, Saslow and Wood [49], who reported adherence rates of over two years for self-prescribed LCDs. This finding challenges the idea that dietary compliance beyond six months is unlikely on a very low-carbohydrate diet [37]. The majority of participants also experienced a reduction in body weight alongside other unanticipated positive physical changes [20–24]. A further health benefit report due to the LCD included an increase in energy. This supports prior research describing more significant energy expenditure during weight loss maintenance from a LCD than a diet high in carbohydrates [18]. Brown [50] has also highlighted how individuals following a LCD are likely to feel 'euphoric', which reflects the increased positivity experienced by participants in the current study.

Physical Opportunity and Social Opportunity for behaviour change can be considered the most challenging components for these individuals to have adjusted to. Social feedback from health professionals and their social network appeared to greatly influence their perceived ability to commence and continue adherence to a LCD. All participants reported negative feedback, challenges, concern from significant others or a lack of general support at some point. These significant others, included health professionals such as the participant's primary care doctor, or diabetes health professional [39], and family members or close friends. The participants in this study reported being able to successfully educate and communicate with their significant others overtime (including health professionals), to gain support or acceptance of their LCD, and this appeared to be a key influence in their success of implementing a LCD long-term. Regardless of social support from close friends and family, all participants acknowledged the online LCD community as a source of opportunity and reinforcement. The challenges associated with Opportunity as a component to change diet behaviour may help explain

why some people cannot sustain the application of a LCD over time [37]. Research highlights that sufficient social support plays an important role in diabetes self-care behaviours and has even been associated with greater glycaemic control [51]. It might be helpful for resources to be developed that focus on the role of social support and communication from both health professionals and significant others in the long-term adoption of a LCD.

## Strengths and limitations

To our knowledge, this is one of the first studies to qualitatively examine the experience of individuals with T2DM adhering to a self-prescribed LCD. The findings sought to utilise psychological investigation to understand lay people's perceptions of LCDs and their reasoning and motivations for engaging in and sustaining such a diet. The study also explores participant's perceived effects of LCD's on their T2DM and general well-being and therefore offers insight for implications to clinical practice. Individuals were recruited through an UK online diabetes community and recruited individuals who had sustained a LCD over a significant period. Further investigation with newly diagnosed patients with T2DM who are exploring dietary control for T2DM or those new to adopting a LCD should be the target of future research on this topic. The findings relate only to those living with T2DM and do not transfer to those with other forms of diabetes, and the findings may not generalise to those beyond the UK healthcare system. As a qualitative study, the methodology did not seek to quantify the participants LCD or to assess changes in diabetes clinical outcomes (e.g. HbA1c), rather it sought to explore their perceptions and experiences of choosing a LCD. Further research to explore the implementation of LCD in practice, and to assess changes in clinical outcomes for diabetes is warranted. Finally, this study is not advocating nor dismissing a LCD for those with T2DM. However, this research simply explores the adoption of and adherence to LCD for this patient group.

## Implications for practice

This observational study is a valuable addition to the low-carb literature. Findings suggest a need for health professionals to engage in open discussions with patients regarding their preferred T2DM treatment approach. This study further highlights the importance of social acceptance and social support to help individuals engage with and maintain a LCD. Health professionals should be actively updated on the position statements referring to dietary advice for those living with T2DM [10, 31–33, 52]. Resources and interventions should be developed to guide health professionals in the communication and exploration of personalised dietary choices to manage T2DM; evidence-based fact files would be a helpful starting point and could be used to explore the knowledge of LCD, and acknowledge the skills required to engage in a LCD. Moreover, patients may benefit from resources tailored towards communicating with friends and family about their adoption of a LCD.

## Conclusion

This is the first qualitative study to explore participant's experiences of utilising a Low-Carbohydrate diet for people living with Type 2 Diabetes. The findings are discussed with reference to a psychological model of behaviour, *COM*-B. Participants reported gaining knowledge and skills to increase their *C*apability to engage in LCDs, *M*otivation to manage diabetes outcomes influenced adherence. However, challenges were reported with the *O*pportunity to engage in behaviour, mainly influenced by social support. Health professionals and significant others may benefit from resources to help build knowledge and understanding and assist with maintaining a LCD long-term.

## Acknowledgments

The authors would like to thank the participants who took part in this study and diabetes.co. uk acting as gatekeepers advertising the study on their website.

## Author Contributions

**Conceptualization:** Lisa Newson, Francesca Helen Parody.

**Data curation:** Lisa Newson, Francesca Helen Parody.

**Formal analysis:** Lisa Newson, Francesca Helen Parody.

**Investigation:** Lisa Newson, Francesca Helen Parody.

**Methodology:** Lisa Newson, Francesca Helen Parody.

**Project administration:** Lisa Newson.

**Supervision:** Lisa Newson.

**Validation:** Lisa Newson, Francesca Helen Parody.

**Visualization:** Lisa Newson, Francesca Helen Parody.

**Writing – original draft:** Francesca Helen Parody.

**Writing – review & editing:** Lisa Newson.

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
