## [Decision Letter · Decision Letter 0]

23 Jun 2022

PONE-D-22-06248Investigating the experiences of low-carbohydrate diets for people living with Type 2 Diabetes: A Thematic AnalysisPLOS ONE

Dear Dr. Newson,

Thank you for submitting your manuscript to PLOS ONE. After careful consideration, we feel that it has merit but does not fully meet PLOS ONE’s publication criteria as it currently stands. Therefore, we invite you to submit a revised version of the manuscript that addresses the points raised during the review process.

The current manuscript outlined the experiences of low-carbohydrate diets for people living with Type 2 Diabetes. The research interest if good. However, some important bottlenecks are pending to be improved to further clarify the results justifications. Kindly take note that a human ethical approval from institutional review board is mandatory (authors are required to provide the approval number). Also, the manuscript is recommended to add elements such as strengths and limitations.==============================

We look forward to receiving your revised manuscript.

Kind regards,

Lai Kuan Lee

Academic Editor

PLOS ONE

Journal Requirements:

2. "Peer review at PLOS ONE is not double-blinded (https://journals.plos.org/plosone/s/editorial-and-peer-review-process). For this reason, authors should include in the revised manuscript all the information removed for blind review

Reviewers' comments:

Reviewer's Responses to Questions

**Comments to the Author**

1. Is the manuscript technically sound, and do the data support the conclusions?

Reviewer #1: Yes

Reviewer #2: Yes

Reviewer #3: Yes

2. Has the statistical analysis been performed appropriately and rigorously? 

Reviewer #1: N/A

Reviewer #2: N/A

Reviewer #3: Yes

3. Have the authors made all data underlying the findings in their manuscript fully available?

Reviewer #1: Yes

Reviewer #2: No

Reviewer #3: Yes

4. Is the manuscript presented in an intelligible fashion and written in standard English?

Reviewer #1: Yes

Reviewer #2: Yes

Reviewer #3: Yes

5. Review Comments to the Author

Reviewer #1: Type 2 diabetes negatively affects the mental health as well as the physical health of the individual, especially in the long term. We need to learn how patients feel about their illness and care about it. The number of such studies should definitely be increased. I congratulate you for your great work !

Reviewer #2: 1. As it is qualitative research kindly justify how can you ensure participants gave proper answer your online interview

2. As you mentioned in the theme-I lack of professional guidance for dietary management of T2DM, How can you ensure they following LCD diets and have you calculated their carbohydrates intake or study participants knows carbohydrates counting.

3. As you mentioned in the theme-IV LCD had improved their T2DM and HbA1c dropped significantly. have you analyzed their pre post HbA1c level and other blood profiles

Reviewer #3: Newson and Parody sought to explore the experiences of participants living with Type 2 Diabetes, utilizing a Low-Carbohydrate diet. Participants reported gaining knowledge and skills to increase their Capability to engage in LCDs. Motivation to manage diabetes outcomes influenced adherence. However, challenges were reported with the Opportunity to engage in behavior, mainly influenced by social support. The manuscript was generally technically sound, and the data support the conclusions. The thematic analysis was very well outlined and very coherent. The written English was clear and very standard.

Strengths of the study. The study area is an emerging one with rising interest. This study serves as a useful baseline study that sought to utilize a psychological investigation to understand lay people's perceptions of LCDs and their reasoning and motivations for engaging in such a diet. Also, the research also focused on how people could sustain their LCD, how they managed it, and also explored potential barriers or challenges they faced. In addition, the study also explored the perceived effects of LCDs on people's T2DM and their general health and well-being.

Limitations of the study were well outlined by the authors.

General comments

1. Ethics approval number and the institution that granted the ethics were not indicated--- line 180

2. Figure 1 quality appears to be low. Authors should submit high resolution figure if manuscript is accepted eventually.

Other corrections and suggestions have been uploaded

6. PLOS authors have the option to publish the peer review history of their article (what does this mean?). If published, this will include your full peer review and any attached files.

Reviewer #1: No

Reviewer #2: No

Reviewer #3: **Yes: **Frank Ekow Atta Hayford

---

## [Author Response · Author response to Decision Letter 0]

5 Jul 2022

Dear Editor and reviewers

Thank you for your helpful and constructive comments, greatly appreciated. We have responded to these as follows:

Editor#: The current manuscript outlined the experiences of low-carbohydrate diets for people living with Type 2 Diabetes. The research interest if good. However, some important bottlenecks are pending to be improved to further clarify the results justifications. Kindly take note that a human ethical approval from institutional review board is mandatory (authors are required to provide the approval number). Also, the manuscript is recommended to add elements such as strengths and limitations.

Thank you, we have responded to the reviewer specific points below. Specifically we have responded to the comments on the ‘quantitative’ based queries and have added these in to the limitations of the study, and we have amended some language in the findings to clarify these are self-reported clinical improvements in line with qualitative methodology. We have clarified the ethical approval within the manuscript and added strengths into the now strengths and limitations section of the discussion. We have reviewed the quality of figure 1 (and checked this using PACE tool) and we have checked the format of the manuscript against the journal guidelines. We have un-blinded the manuscript and added author initials etc. where appropriate. 

Reviewer #1: Type 2 diabetes negatively affects the mental health as well as the physical health of the individual, especially in the long term. We need to learn how patients feel about their illness and care about it. The number of such studies should definitely be increased. I congratulate you for your great work !

Thank you for your comments. 

Reviewer #2: 1. As it is qualitative research kindly justify how can you ensure participants gave proper answer your online interview

Thank you for your comment. We believe we have provided a detailed description of our data collection process in line with qualitative, thematic analysis methodology. As describe the researcher built rapport with the participants and then the interviews were recorded, and later transcribed. Participants were given every opportunity discuss and answer questions, as described within the methodology:

“Semi-structured interviews allowed for detailed and personal discussions surrounding the participants' experience of LCDs. The use of open-ended questions such as "How have you found maintaining your LCD?" gave the participants the opportunity to discuss aspects important to them. This also allowed the researcher to follow up any themes of interest that developed from the discussions. An interview schedule was used as a basis to guide the semi-structured interviews”. 

“ Prompt questions were also included to ensure that the interview imitated a 'flowing conversation' [44].”

Reviewer #2.

2. As you mentioned in the theme-I lack of professional guidance for dietary management of T2DM, How can you ensure they following LCD diets and have you calculated their carbohydrates intake or study participants knows carbohydrates counting.

3. As you mentioned in the theme-IV LCD had improved their T2DM and HbA1c dropped significantly. have you analyzed their pre post HbA1c level and other blood profiles

We have clarified the language in the theme 4 to be more specific that these changes were self-reported. 

For example “All respondents reported that their HbA1c levels had dropped significantly”

Regards points 2 & 3- we have added the following into the strengths and limitations section of the discussion to acknowledge these points:

“As a qualitative study, the methodology did not seek to quantify the participants LCD or to assess changes in diabetes clinical outcomes (e.g. HbA1c), rather it sought to explore their perceptions and experiences of choosing a LCD. Further research to explore the implementation of LCD in practice, and to assess changes in clinical outcomes for diabetes is warranted.”

Reviewer #3: Newson and Parody sought to explore the experiences of participants living with Type 2 Diabetes, utilizing a Low-Carbohydrate diet. Participants reported gaining knowledge and skills to increase their Capability to engage in LCDs. Motivation to manage diabetes outcomes influenced adherence. However, challenges were reported with the Opportunity to engage in behavior, mainly influenced by social support. The manuscript was generally technically sound, and the data support the conclusions. The thematic analysis was very well outlined and very coherent. The written English was clear and very standard.

Thank you for these comments

Strengths of the study. The study area is an emerging one with rising interest. This study serves as a useful baseline study that sought to utilize a psychological investigation to understand lay people's perceptions of LCDs and their reasoning and motivations for engaging in such a diet. Also, the research also focused on how people could sustain their LCD, how they managed it, and also explored potential barriers or challenges they faced. In addition, the study also explored the perceived effects of LCDs on people's T2DM and their general health and well-being.

We have added a strengths sections specifically into the discussion, and utilised your feedback to assist with this. 

Limitations of the study were well outlined by the authors.

Thank you

General comments

1. Ethics approval number and the institution that granted the ethics were not indicated--- line 180

Ethical approval details have now been confirmed within the manuscript itself.

2. Figure 1 quality appears to be low. Authors should submit high resolution figure if manuscript is accepted eventually. 

We have reviewed the Figure 1 quality again, thank you. We have checked this with PACE tool for quality, thank you.

Other corrections and suggestions have been uploaded-

Thank you, we have made the four minor amends to the transcript as suggested via the attached upload. 

We hope this satisfies the review and welcome your response. 

Many thanks

Dr Newson

---

## [Decision Letter · Decision Letter 1]

9 Aug 2022

Investigating the experiences of low-carbohydrate diets for people living with Type 2 Diabetes: A Thematic Analysis

PONE-D-22-06248R1

Dear Dr. Newson,

We’re pleased to inform you that your manuscript has been judged scientifically suitable for publication and will be formally accepted for publication once it meets all outstanding technical requirements.

Kind regards,

Lai Kuan Lee

Academic Editor

PLOS ONE

Additional Editor Comments (optional):

Reviewers' comments:

Reviewer's Responses to Questions

**Comments to the Author**

1. If the authors have adequately addressed your comments raised in a previous round of review and you feel that this manuscript is now acceptable for publication, you may indicate that here to bypass the “Comments to the Author” section, enter your conflict of interest statement in the “Confidential to Editor” section, and submit your "Accept" recommendation.

Reviewer #3: All comments have been addressed

2. Is the manuscript technically sound, and do the data support the conclusions?

Reviewer #3: Yes

3. Has the statistical analysis been performed appropriately and rigorously? 

Reviewer #3: Yes

4. Have the authors made all data underlying the findings in their manuscript fully available?

Reviewer #3: Yes

5. Is the manuscript presented in an intelligible fashion and written in standard English?

Reviewer #3: Yes

6. Review Comments to the Author

Reviewer #3: The authors have adequately addressed all the comments raised by reviewers. Authors should ensure that guidelines for the journal is strictly followed

7. PLOS authors have the option to publish the peer review history of their article (what does this mean?). If published, this will include your full peer review and any attached files.

Reviewer #3: **Yes: **Dr Frank Ekow Atta Hayford

---

## [Editor Report · Acceptance letter]

12 Aug 2022

PONE-D-22-06248R1 

Investigating the experiences of low-carbohydrate diets for people living with Type 2 Diabetes: A thematic analysis. 

Dear Dr. Newson:

I'm pleased to inform you that your manuscript has been deemed suitable for publication in PLOS ONE. Congratulations! Your manuscript is now with our production department. 

Kind regards, 

on behalf of

Dr. Lai Kuan Lee 

Academic Editor

PLOS ONE